# AUTOREGRESSIVE VIDEO GENERATION BEYOND NEXT FRAMES PREDICTION

## ABSTRACT

Autoregressive models for video generation typically operate frame-by-frame, extending next-token prediction from language to video's temporal dimension. We question that unlike word as token is universally agreed in language if frame is a appropriate prediction unit? To address this, we present VideoAR, a unified framework that supports a spectrum of prediction units including full frames, key-detail frames, multiscale refinements, and spatiotemporal cubes. Among these designs, we find model video generation using *spatiotemporal* cubes as prediction units, which allows autoregressive models to operate across both spatial and temporal dimensions simultaneously. This approach eliminates the assumption that frames are the natural atomic units for video autoregression. We evaluate VideoAR across diverse prediction strategies, finding that cube-based prediction consistently delivers superior quality, speed, and temporal coherence. By removing the frame-by-frame constraint, our video generator surpasses state-of-the-art baselines on VBench while achieving faster inference and enabling seamless scaling to minute-long sequences. We hope this work will motivate rethinking sequence decomposition in video and other spatiotemporal domains.

## 1 INTRODUCTION

In recent years, video generation (Brooks et al., 2024; Polyak et al., 2024; Kondratyuk et al., 2023; Zheng et al., 2024; Agarwal et al., 2025; Wan et al., 2025; Kong et al., 2024) has made remarkable strides, which iteratively denoise entire sequences via full-attention mechanisms. While achieving impressive quality, these methods suffer from quadratic scaling in both computation and memory, making high-resolution or long-duration videos prohibitively expensive.

Autoregressive models have emerged as a promising alternative (Yin et al., 2024b; Zhang & Agrawala, 2025; Alonso et al., 2024; Jin et al., 2024), decomposing video generation into sequential predictions that avoid this quadratic bottleneck. Following the success of autoregression in language modeling (Achiam et al., 2023; Touvron et al., 2023), current approaches have naturally adopted frame-by-frame prediction: generating videos one frame at a time, with each frame conditioned on all previously synthesized frames. This prevailing strategy has led to a widespread belief that autoregressive video generation is inherently linked to frame-level decomposition that frames are the natural "tokens" of video, analogous to words in language.

In this work, we aim to address the following question: *Can autoregressive video generation be generalized to any prediction unit? Which prediction unit yields the best performance?* We note that unlike language, where tokens (words) form a universally agreed-upon unit, the fundamental prediction unit in video remains an open question. Videos are inherently spatiotemporal, and decomposing them purely along the temporal dimension may not capture their true structure. This intuition is supported by human vision research (Yarbus, 2013; Dorr et al., 2010), which shows that humans explore dynamic scenes through region-based scans rather than exhaustive frame-by-frame processing. The autoregressive principle that predicts next units based on previous ones is agnostic to the choice of unit itself.

With this observation, we propose that autoregressive video generation can operate on more flexible, semantically meaningful units that span both spatial and temporal dimensions. We introduce VideoAR, a generalized framework that can be instantiated with various prediction granularities: frames, key-detail hierarchies, spatiotemporal cubes, and multi-scale pyramids.

Through systematic evaluation, we discover that *spatiotemporal cubes* that extend across both space and time consistently deliver superior generation quality, speed, and temporal coherence.

Our cube-based approach addresses fundamental limitations of frame-by-frame prediction. By conditioning on units that capture local appearance and motion jointly, we minimize the exposure bias inherent in sequential frame generation, where errors compound along the temporal axis. Moreover, this design enables VideoAR to scale from five-second clips to ultra-long sequences spanning multiple scenes without requiring long video training data that full-attention models depend on. Each predictive cube propagates information along both spatial and temporal axes, in contrast to frame-based schemes that confine autoregressive progression to the temporal dimension alone. As shown in Figure 1, our VideoAR outperforms previous diffusion based and autoregressive model.

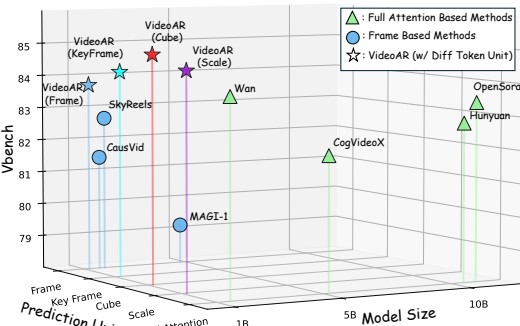

Figure 1: Comparison with open source model on VBench. Our VideoAR outperforms all diffusion models and autoregressive models.

Experimental results demonstrate that VideoAR surpasses state-of-the-art diffusion and frame-based autoregressive baselines on VBench (Huang et al., 2024) and human evaluation, while achieving significantly faster inference. The effectiveness of spatiotemporal prediction units reveals a largely unexplored design space in video generation: treating video as a collection of spatiotemporal cubes rather than a sequence of frames. Given the efficiency, quality, and scalability of our method, we believe VideoAR represents a fundamental shift in how autoregressive models approach video generation, moving beyond the constraints of next-frame prediction toward more flexible and efficient decompositions.

## 2 RELATED WORK

We review two dominant paradigms for video generation in this section including diffusion-based methods in Sec. 2.1 and autoregressive video generation in Sec. 2.2.

### 2.1 DIFFUSION FOR VIDEO GENERATION

Early work such as Video Diffusion Models (Ho et al., 2022b) extended image denoising diffusion to short clips by applying 3D U-Nets over space-time volumes, achieving coherent $64{\times}64$ videos at the cost of heavy computation. Cascaded or latent approaches subsequently pushed both quality and duration: Imagen Video (Ho et al., 2022a) chains spatial and temporal superresolution models to reach $1280{\times}768$ at 24 fps for $\sim$5 s clips, while Phenaki (Villegas et al., 2022) employs a compressed latent representation to generate minute long, story conditioned sequences. Make-A-Video (Singer et al., 2022) and ModelScope T2V (Wang et al., 2023) demonstrated text-to-video generation with text-to-image backbones adapted to temporal dynamics.

Recent diffusion and flow-matching models have set the standard for video generation quality. DiT-based architectures (Peebles & Xie, 2023) underpin state-of-the-art systems including Sora (Brooks et al., 2024), MovieGen (Polyak et al., 2024), HunyuanVideo (Kong et al., 2024), and Wan (Wan et al., 2025), which leverage spatiotemporal transformers and massive text-video datasets to achieve photorealistic results. Despite impressive fidelity, these full-attention designs face fundamental limitations: quadratic scaling with spatiotemporal resolution, slow sampling speed, prohibitive memory requirements for long sequences.

### 2.2 AUTOREGRESSIVE VIDEO GENERATION

Autoregressive models offer a compelling alternative by decomposing video generation into sequential predictions, naturally aligning with video's temporal structure while avoiding quadratic complexity. Early approaches explored regression (Liu et al., 2017) and adversarial losses (Goodfellow

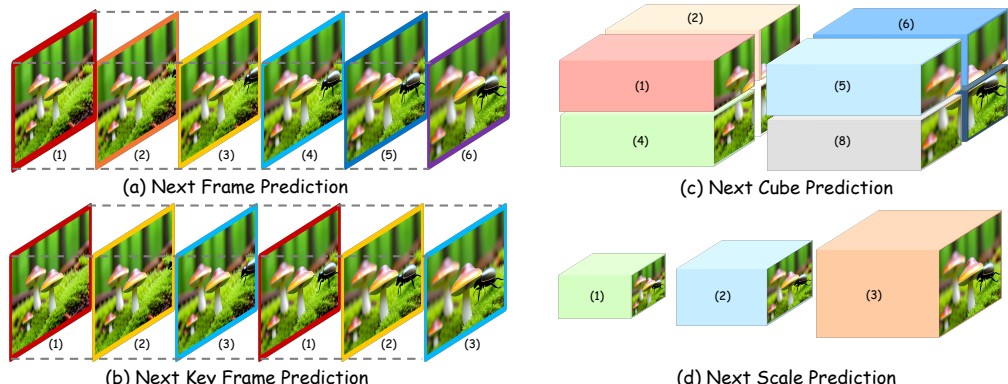

Figure 2: **Generalized Autoregressive Video Generation.** We generalize next-token autoregressive modeling to next any prediction unit autoregressive modeling including (a) a frame (Yin et al., 2024b; Zhang & Agrawala, 2025) (b) strided frame group (grouping non-adjacent frames) (c) spatial-temporal cube. (d) a scale (coarse to fine resolution, inspired by VAR (Tian et al., 2024)). We use different color to represent autoregressive steps.

et al., 2020; Villegas et al., 2017; Vondrick & Torralba, 2017) for frame prediction. Inspired by the success of large language model, VideoGPT (Yan et al., 2021) and DVD-GAN (Clark et al., 2019) quantize frames into discrete tokens and apply next-token prediction, though raster-order generation proved inefficient. Recent architectures address these limitations through different strategies: VideoPoet (Kondratyuk et al., 2023) unifies multimodal generation through a decoder-only transformer, Cosmos (Agarwal et al., 2025) frames world modeling as video generation for embodied AI, and FramePack (Zhang & Agrawala, 2025) combines frame compression with anti-drifting sampling to reduce error accumulation.

However, current autoregressive approaches universally adopt frame-by-frame prediction. CausVid (Yin et al., 2024b), the most relevant baseline, generates one complete frame at a time—limiting spatial context and causing errors to compound over long sequences. This frame-level constraint reflects an unexamined assumption that frames are the natural units for video autoregression. In contrast, VideoAR introduces a generalized framework supporting flexible prediction units: frames, key-detail hierarchies, spatiotemporal cubes, and multiscale refinements. Our systematic evaluation reveals that cube-based prediction, jointly modeling space and time, substantially improves quality, coherence, and efficiency while preserving computational advantages.

## 3 METHOD

We first provide a formal autoregressive framework for video generation in Sec. 3.1. In Sec. 3.2, we explore different prediction units. In Sec. 3.3, we propose the generalized autoregressive video generation equipped with symmetry distribution matching distillation. Finally, we discuss how videoAR generate long videos without long video as training data in Sec. 3.4.

### 3.1 GENERALIZED AUTOREGRESSIVE FRAMEWORK FOR VIDEO

Unlike language where tokens form natural units, video can be decomposed into various prediction granularities. We propose VideoAR, a unified framework that generalizes autoregressive modeling beyond frame-by-frame prediction. Given a video latent $z \in \mathbb{R}^{T \times H \times W \times C}$ from the video encoded by a pretrained VAE (Wan et al., 2025; Kingma & Welling, 2013), we partition it into an ordered sequence of units $\mathcal{X} = \{X_1, X_2, \ldots, X_N\}$ and model the factorization:

$$p_\theta(\mathcal{X}) = \prod_{i=1}^{N} p_\theta(X_i \mid X_{<i}) \tag{1}$$

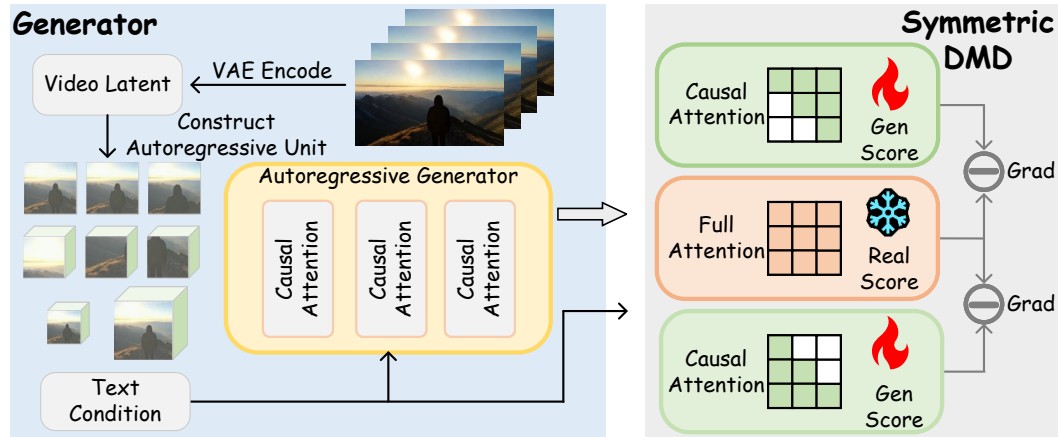

Figure 3: **Generalized Autoregressive Video Generation.** The video latent from the video encoded by VAE encoder is first constructed into prediction units and the autoregressive generator causally generate videos supervised by symmetric distribution matching loss. Different from vanilla AR which uses cross entropy loss on each token. We design symmetric distribution matching loss on each prediction unit. We use base model with full attention as real score function and two causal model with symmetric attention mask together as generated data score function.

The key insight is that the unit $X_i$ need not be a single frame. By varying the unit definition, we control the trade-off between context richness, computational efficiency, and error accumulation.

## 3.2 PREDICTION UNIT DESIGN SPACE

We systematically explore four unit types, each with distinct computational and modeling trade-offs:

**Frame.** Each unit $X_i \in \mathbb{R}^{H \times W}$ represents one complete frame, yielding $N = T$ autoregressive steps. The model predicts each frame conditioned on all previous frames, with information flowing strictly along the temporal axis. While intuitive, this provides minimal context per prediction and compounds errors over long sequences.

**Key-Detail Frame.** Inspired by video compression schemes (Wiegand et al., 2003) that store periodic reference frames first and the rest frames next, we first predict key frames at regular intervals, then fill intermediate frames. Each unit $X_i \in \mathbb{R}^{\frac{T}{k} \times H \times W}$ contains $\frac{T}{k}$ temporally-sparse frames, reducing autoregressive steps to $N = k$. This hierarchical approach exposes longer-range motion patterns while maintaining spatial completeness.

**Spatiotemporal Cube.** Each unit $X_i \in \mathbb{R}^{k_t \times k_h \times k_w}$ forms a 3D cube spanning $k_t$ frames and a $k_h \times k_w$ spatial patch, yielding $N = \frac{T}{k_t} \cdot \frac{H}{k_h} \cdot \frac{W}{k_w}$ steps. Unlike frames that restrict causality to time, cubes propagate information along both spatial and temporal axes. This joint encoding captures local appearance and motion within each unit, reducing error accumulation while providing richer context – analogous to how humans perceive video through region-based scanning rather than frame-by-frame processing.

**Multiscale.** Following VAR (Tian et al., 2024), we construct a pyramid $\{\mathbf{z}^{(1)}, \ldots, \mathbf{z}^{(L)}\}$ via downsampling. We define $\mathbf{z}^i = \mathrm{resize}(\mathbf{z}, s_i)$, where resize refer to downsample $\mathbf{z}$ to target scale $s_i$. Therefore, $\mathbf{z}^{(1)}$ is the coarsest scale and $\mathbf{z}^{(L)}$ the finest scale, namely $\mathbf{z}$. Each scale forms one prediction unit, resulting in just $N = L$ steps. The model learns $p(\mathbf{z}^{(\ell)} \mid \mathbf{z}^{(<\ell)})$, progressively refining from global structure to fine details. Early units are drastically smaller, enabling efficient coarse-level generation before detail refinement.

This unified framework enables systematic comparison of how unit choice affects generation quality, speed, and temporal coherence in autoregressive video modeling.

### 3.3 GENERALIZED AUTOREGRESSIVE VIDEO GENERATION

As shown in Figure 3. a VAE encodes a video into a sequence of latent prediction units (e.g., frames, scales or spatiotemporal cubes). A autoregressive generator with stacked causal attention generate each next unit, conditioned on the text prompt and previously generated units. At inference, only this generator is used and the VAE decoder maps latents back to pixels.

Standard autoregressive generators (Touvron et al., 2023; Sun et al., 2024; Achiam et al., 2023; Yan et al., 2021) over discrete tokens are trained with a token-wise cross-entropy objective. In contrast, our generator operates in a continuous latent space and advances in various prediction units, which calls for a different training signal. We propose Symmetric Distribution Matching Distillation (Symmetric DMD), a novel training framework inspired by DMD (Yin et al., 2024a) which aligns the generated data distribution $p_{\text{gen}}$ and the real data distribution $p_{\text{data}}$. This is achieved by minimizing the reverse KL-divergence, for which the loss gradient can be approximated as:

$$\mathcal{L} \approx \mathbb{E}_z \left[ \mathbb{E}_{\tilde{x} \sim q(\cdot | G_\theta(z))} \big[ s_{\text{real}}(\tilde{x}) - s_{\text{gen}}(\tilde{x}) \big] \nabla_\theta G_\theta(z) \right] \tag{2}$$

where $z$ is sampled from a noise prior, $G_\theta$ is the generator, and $s_{\text{real}}, s_{\text{gen}}$ are real and generated data score functions. DMD use the model with full attention as $s_{\text{real}}$ and $s_{\text{gen}}$.

**Symmetry Distribution Matching Distillation.** A critical weakness in applying standard DMD to VideoAR is the mismatch between a causal generator and non-causal generated data score function. Our Symmetric DMD resolves this by keeping real score as bidirectional scorer and defining two distinct generated data scorers with specific inductive biases to align with generator's casualty and real score's capacity.

For Real Scorer ($s_{\text{real}}$), to accurately model the complex distribution of real videos, we use a powerful score model with full attention. This model can process the entire video sequence simultaneously. This model is pretrained on real data and frozen during generator training (see Fig. 3, snowflake icon).

For Causal Generated Scorer ($s_{\text{gen}}$), to respect causality alignment with the generator and to match capacity with real scorer, we instantiate the fake score as an architectural replica of the generator and use two such scorers with symmetric causal masks: one forward and one backward. Each scorer is strictly causal in its respective direction, preserving the generator's inductive bias; taken together, the pair covers both temporal directions and narrows the capacity gap to the full-attention real scorer. The model is finetuned on generated data (see Fig. 3, fire icon).

### 3.4 ZERO-SHOT LONG VIDEO GENERATION

Diffusion-based video generators are costly to scale to long durations: they denoise all frames jointly for dozens of steps, so compute and memory grow with the number of frames per sample. In addition, models trained on fixed-length clips often fail to extrapolate when asked to produce much longer videos, the denoising schedule and receptive field no longer match the target horizon, leading to drift and temporal incoherence. Besides, the difficulty of collecting long video as training data also impede the long video generation with diffusion model.

VideoAR overcomes these limitations with a streaming autoregressive formulation. Instead of denoising an entire clip at once, VideoAR generates a sequence of units, one at a time, each conditioned on previously generated units. This design enables zero-shot long video generation without any long video as training data. At inference, we reuse the keys/values from earlier units via a KV cache and gradually drops earlier KV cache to keep the overall sequence length short. The cost of producing the next unit is constant w.r.t. a predefined video length. Therefore, the compute and memory scale with the chosen context length, not with the entire sequence. This makes, in principle, arbitrarily long ("infinite") videos feasible. Video use flexible context to balance coherence and efficiency. The context (cache) length is an inference-time knob, we can utilize fixable context length based on computation resources.

We adopt next cube prediction as default settings of our VideoAR. Next-cube factorization reduces error accumulation, because each unit carries joint *spatio–temporal* information and the information is propagated both across space and time. Compared to next-frame predictors, the next-cube design provides thicker temporal context per decision and stronger spatial coupling, which lowers exposure-

Table 1: **Quantitative Comparison.** We compare our VideoAR with representative open-source diffusion and autoregressive video generation models under similar parameters.

| Model | #Params | Throughput (FPS) ↑ | Vbench Scores ↑ | | |
|---|---|---|---|---|---|
| | | | Total Score | Quality | Semantic |
| *Diffusion models* | | | | | |
| LTX-Video (HaCohen et al., 2024) | 1.9B | 8.98 | 80.15 | 82.52 | 70.68 |
| CogVideoX (Yang et al., 2024) | 5B | 0.85 | 82.04 | 82.68 | 79.48 |
| OpenSora (Zheng et al., 2024) | 11B | 0.08 | 83.40 | 84.28 | 79.89 |
| HunyuanVideo (Kong et al., 2024) | 13B | 0.22 | 82.80 | 84.64 | 75.42 |
| Wan2.1 (Wan et al., 2025) | 1.3B | 0.78 | 83.88 | 85.02 | 79.32 |
| *Autoregressive models* | | | | | |
| SkyReels-V2 (Chen et al., 2025) | 1.3B | 0.49 | 82.67 | 84.70 | 74.53 |
| MAGI-1 (Teng et al., 2025) | 4.5B | 0.19 | 79.18 | 82.04 | 67.74 |
| NOVA (Deng et al., 2024) | 0.6B | 0.88 | 80.12 | 80.39 | 79.05 |
| Pyramid Flow (Jin et al., 2024) | 2B | 6.7 | 81.72 | 84.74 | 69.62 |
| CausVid (Yin et al., 2024b) | 1.3B | 17.0 | 81.46 | 84.05 | 69.80 |
| VideoAR (Ours) | 1.3B | 16.4 | 84.87 | 85.92 | 80.67 |

bias accumulation and preserves dynamics over minutes of generation without any training on long videos.

## 4 EXPERIMENTS

In this section, we present implementation details and evaluation metrics first. In Sec. 4.1, we present quantitative results of text to image generation including automatic evaluation and user study. In Sec. 4.2, we show qualitative results including short and long video generation. Finnally, we present ablation study in Sec. 4.3

**Implementation details.** We implement our VideoAR based on Wan2.1-1.3B (Wan et al., 2025), a flow matching (Lipman et al., 2022) based model. The model training are divided into two stages: 1) finetuning base model with causal mask. We use 100k data sampled from base model. The optimizer is AdamW with the learning rate of 1e-4. This stage aims to recover the causal generation ability of our autoregressive model. We train 30k iterations in this stage. 2) finetuning the model from stage one with internal dataset and the proposed symmetric DMD loss. The learning rate is set to 2e-6. We train 6k iterations in this stage.

**Evaluation metric.** We adopt VBench (Huang et al., 2024) with 946 prompts including 16 sub-metrics as automatic metric to evaluate both visual quality and semantic alignment. We also sample videos for user preference study.

### 4.1 TEXT TO IMAGE GENERATION

For a fair comparison, we evaluate VideoAR against representative diffusion and autoregressive video generators, reporting model parameters, inference throughput in frames per second (FPS) and VBench total scores seen in Table 1.

Compared with diffusion models, VideoAR attains the highest VBench total score among all models, reaching 84.87, outperforming strong diffusion baselines such as Wan2.1 (+ 0.99), OpenSora (+ 1.47), HunyuanVideo (+ 2.07), and LTX-Video (+ 4.72). At the same time, VideoAR is substantially faster at inference: 16.4 FPS versus 0.78 (Wan2.1; $\sim 21 \times$).

Compared with autoregressive models, VideoAR delivers the best VBench Total score (84.87), exceeding CausVid (+3.41), SkyReels-V2 (+2.20), NOVA (+4.75 ), MAGI-1 (+5.69), and Pyramid Flow (+3.15). In terms of speed, VideoAR achieves 16.4 FPS, on par with the fastest frame-wise autoregressive model (CausVid, 17.0 FPS), while being markedly faster than the rest (e.g., $\sim 2.5 \times$ vs. Pyramid Flow at 6.7 FPS, $\sim 33 \times$ vs. SkyReels-V2 at 0.49 FPS, and $\sim 86 \times$ vs. MAGI-1 at 0.19 FPS). Overall, our VideoAR establishes the best quality and throughput. it sets the top VBench Total

Table 2: **Ablation on Prediction Unit.** Under the same prediction unit (frame), our VideoAR outperforms CausVid. Among different prediction unit, using cube as prediction unit shows best performance.

| Method | Prediction Unit | Total Score |
|---|---|---|
| CausVid (Yin et al., 2024b) | frame | 81.46 |
| VideoAR | frame | 83.72 |
| VideoAR | key-detail frame | 84.24 |
| VideoAR | Scale | 84.51 |
| VideoAR | cube | 84.87 |

Table 3: **Ablation on Cube Size.** Varying the spatiotemporal cube size

| Cube Size | Number of Unit | Total Score |
|---|---|---|
| $\frac{t}{4} \times \frac{h}{4} \times \frac{w}{4}$ | $4 \times 4 \times 4$ | 84.22 |
| $\frac{t}{2} \times \frac{h}{2} \times \frac{w}{2}$ | $2 \times 2 \times 2$ | 84.87 |
| $t \times \frac{h}{2} \times \frac{w}{2}$ | $1 \times 2 \times 2$ | 84.69 |

score across all compared methods while retaining a near state-of-the-art autoregressive decoding speed and delivering orders-of-magnitude faster inference than diffusion-based generators.

**User Study.** As shown in Figure 4, participants consistently preferred our VideoAR over competing approaches in all four pairwise comparisons: 64.3% vs. Wan, 65.6% vs. CausVid, 55.1% vs. MAGI-1, and 68.3% vs. PyramidFlow. All rates exceed the 50% parity line, with an average preference of 63.3% (min 55.1%, max 68.3%), indicating that our method is more appealing than both diffusion and autoregressive baselines.

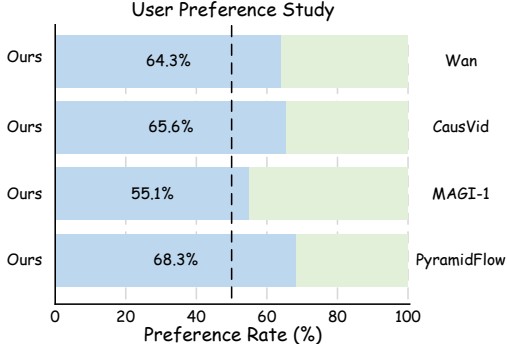

Figure 4: **User Study.** Our VideoAR model outperforms diffusion and autoregressive models in the user preference study.

## 4.2 QUALITATIVE RESULTS.

As shown in Figure 5, we provide qualitative results. Compared with Wan, our VideoAR generates videos with better background consistency, higher frame-wise quality, clearer foreground, and better motions. With higher quality, our VideoAR can generate videos 21.0× faster than Wan.

**Zero-shot Long Video Generation.** Without any long videos as training data, our VideoAR can zero-shot generate extremely long video. As shown in Figure 6, Wan is only able to generate video about 10s, while our VideoAR can generate extremely long video.

## 4.3 ABLATION STUDY

In this section, we conduction ablation study prediction unit design, cube size and our proposed symmetric distribution matching loss.

**Prediction Unit.** Table 2 ablates the choice of autoregressive prediction unit. Under the same unit (*frame*), VideoAR surpasses CausVid by +3.41 total score (84.87 vs. 81.46), indicating that our VideoAR improves performance beyond unit selection alone. Within VideoAR, moving from frame to key-detail frame, scale, and cube yields monotonic gains of +0.52, +0.79, and +1.15 over the frame baseline, respectively, with the cube unit achieving the best score of 84.87. We hypothesize that predicting cube increases spatial-temporal context per step and propogate information along both space and time, thereby boosts the performance.

**Prompt:** *High-speed macro of a hummingbird feeding from a flower; wings produce motion-blurred arcs, iridescent feathers shimmer with view-dependent color; dew drops vibrate, pollen particles scatter; creamy background bokeh.*

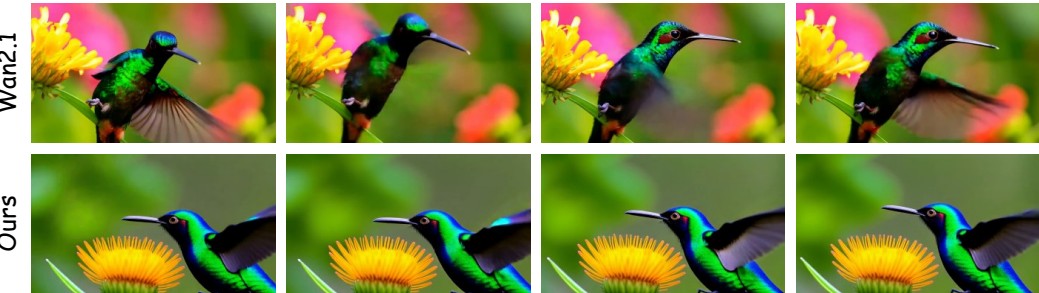

**Prompt:** *Forest floor timelapse-style transition to realtime: mushrooms grow, moss spreads, then a beetle crawls across a glistening leaf; moisture beads, micro-DOF shifts, gentle lens breathing, earthy translucency.*

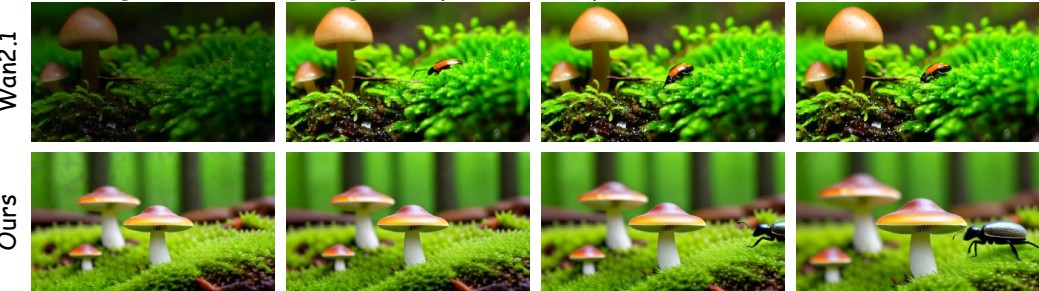

**Prompt:** Train Window Rivulets — Interior of a moving train at night; raindrops on the window merge and trail, creating dynamic bokeh from city lights; camera pans to follow a single droplet down the glass; depth layering shows outside parallax plus interior reflections.

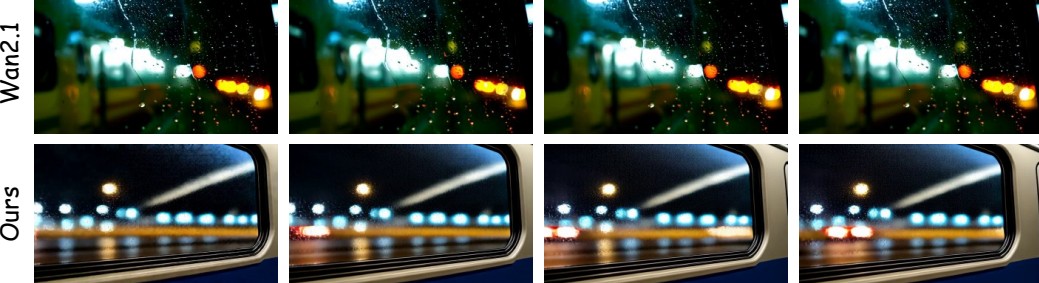

Figure 5: **Qualitative Results.** We visualize video generated by Wan (Wan et al., 2025) and our VideoAR.

Table 4: **Ablation study on Symmetric DMD.** Compared with full attention or vanilla DMD loss, symmetric DMD boost the performance.

| Full | Causal Forward | Causal Backward | Total Score |
|:---:|:---:|:---:|:---:|
| ✓ | | | 84.34 |
| | ✓ | | 84.56 |
| | | ✓ | 84.07 |
| | ✓ | ✓ | 84.87 |

**Cube size.** We study how the prediction unit partitions the spatiotemporal volume into cubes of size $\frac{t}{a} \times \frac{h}{b} \times \frac{w}{c}$ (thus inducing $a \times b \times c$ units). As shown in Table 3, a balanced partition with

**Prompt:** *An underwater dreamscape of neon-glowing jellyfish drifting through a cathedral-like coral reef. Shafts of turquoise solar light pierce the sapphire depths, illuminating swaying anemones. The camera tracks laterally at midwater, capturing the creatures' delicate tendrils and pulsing bioluminescence against the hushed chorus of filtered waves.*

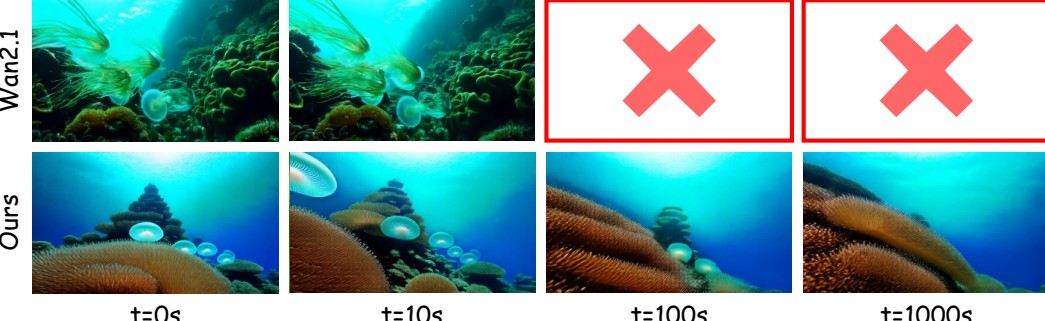

Figure 6: **Long video Generation.** Our videoAR can generate 1000s videos without requiring any long video as training data. Wan (Wan et al., 2025) only support about 10s video and cannot generate such long video.

$\frac{t}{2} \times \frac{h}{2} \times \frac{w}{2}$ (i.e., $2 \times 2 \times 2$ units) yields the best Total Score of 84.87. Using finer cubes $\frac{t}{4} \times \frac{h}{4} \times \frac{w}{4}$ ($4 \times 4 \times 4$ units) degrades performance to 84.22 ($-0.65$), likely due to more autoregressive steps and error accumulation across many inter-cube boundaries. Conversely, enlarging only the temporal span to $t \times \frac{h}{2} \times \frac{w}{2}$ ($1 \times 2 \times 2$ units) slightly underperforms at 83.96 ($-0.18$), suggesting overly coarse temporal granularity within a cube can weaken motion consistency across cube boundaries.

We therefore adopt the $2 \times 2 \times 2$ cube in subsequent experiments.

**Symmetric loss function.** We ablate how the symmetric DMD influence the performance of generator. The core generates ad data score $s_{gen}$ which is parameterized by two causal models with symmetric causal masks. We report the results in Table 4. Using the original full-attention score network from DMD as $s_{gen}$ yields a total score of 84.34. Replacing it with a causal forward mask score which is architecturally identical to our VideoAR, gives better performance, 84.56 (+0.22 over full). In contrast, a causal backward mask (symmetric to causal forward mask) score underperforms at 84.07, and combining forward and backward scores reaches 84.87, which is the best performance. These trends support our design choice: 1) because the generator produces videos autoregressively in a forward order, the generated data score used to model the generated data distribution should respect the same causal constraint. 2) the real and generated data score should be comparable in modeling power to avoid a model capacity gap. We realize this by pairing two causal generated data scorers including one forward, one backward whose combined view approximates the modeling capacity of a full attention real scorer while preserving per-model causality. Therefore, generated data score with causal forward and causal backward yields the best performance.

## 5 CONCLUSION

We presented VideoAR, a generalized autoregressive video generation framework that reframes video generation around the choice of *prediction unit*. Unlike full-attention diffusion models whose cost scales quadratically with spatiotemporal extent, VideoAR shifts computation to sequential prediction and can be instantiated with units ranging from frames and key-detail frames to scales and spatiotemporal cubes. Systematic analysis shows that Next Cube Prediction consistently yields the best quality, speed, and temporal coherence by capturing appearance and local motion within a single unit and reducing error accumulation across long horizons. Besides, our VideoAR shows strong zero-shot long video generation ability. This work highlights prediction granularity as a first-class design axis for autoregressive video models and connects effective units to region-centric patterns observed in human scene exploration. We hope this perspective catalyzes more flexible, efficient, and scalable approaches to video synthesis.

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

# A    APPENDIX

**LLM Usage Declaration.**    We use ChatGPT-5 solely for minor language refinement (e.g., grammar and wording). The tool did not generate original content, arguments, data analysis, or conclusions, nor did it influence the study design or interpretation. All ideas, experiments, and results were conceived and executed without AI assistance.

