# OpenReview forum: "Autoregressive Video Generation beyond Next Frames Prediction"
_ICLR.cc/2026/Conference — ICLR 2026 Conference Withdrawn Submission_

### Official Review · Reviewer_mQYe · 2025-10-27

**Soundness:** 2
**Presentation:** 3
**Contribution:** 2
**Rating:** 2
**Confidence:** 4

**Summary:**

This paper challenges the conventional frame-by-frame paradigm in autoregressive video generation, arguing that the "frame" is not necessarily the optimal prediction unit. It introduces VideoAR, a generalized framework that explores various sequence decomposition strategies. The central claim is that using spatiotemporal cubes as the prediction unit significantly outperforms other units, particularly in generation quality, speed, and temporal coherence. To train this framework, the authors propose Symmetric Distribution Matching Distillation (Symmetric DMD), a knowledge distillation technique that uses a pair of causal scorers to approximate a full-attention teacher model, enabling the autoregressive generator to learn from a powerful pre-trained diffusion model.
Empirically, the method achieves state-of-the-art results on the VBench benchmark for short video generation, even surpassing strong diffusion models in total score while being orders of magnitude faster. The paper further claims a remarkable capability for zero-shot long-video generation, illustrated with an example of a minute-long sequence.

**Strengths:**

1. Addresses a Core Trade-off in Generative Models: The paper tackles the critical and persistent trade-off between generation quality and inference speed in video synthesis. By framing its approach as an autoregressive method distilled from a powerful diffusion model, it directly confronts the challenge of achieving high fidelity without the prohibitive computational cost of iterative denoising, which is a significant and practical research direction.
2. Impressive Inference Speed with Competitive Quality: The most undeniable contribution of this work is its engineering achievement. The proposed VideoAR model achieves an inference speed of 16.4 FPS, which is on par with the fastest existing autoregressive models and over 20 times faster than its state-of-the-art diffusion-based teacher model (Wan2.1). Achieving this level of acceleration while maintaining a VBench score that is not only competitive but marginally higher than the teacher model is a remarkable feat of model compression and distillation.
3. A Systematic, Albeit Exploratory, Study on Prediction Units: The paper introduces a generalized framework for conceptualizing autoregressive video generation beyond the standard per-frame approach. While the foundational claims are debatable, the systematic evaluation of different prediction units (frames, key-frames, scales, and cubes) presented in Table 2 is a valuable exploratory study. It provides empirical evidence suggesting that the choice of the atomic unit for generation is a non-trivial design decision that impacts performance, which could stimulate further investigation in the community.
4. Demonstrates Potential for Long-Sequence Scalability: The autoregressive, streaming nature of the proposed method inherently allows it to scale to sequences of arbitrary length without the quadratic memory and compute explosion seen in full-attention models. While the evidence provided for long-video quality is critically insufficient, the paper does successfully demonstrate the capability of its architecture to maintain a constant generation cost per unit, which is a fundamental prerequisite for any scalable long-video generator.

**Weaknesses:**

1.Fundamental Lack of Novelty and Unclear Contribution: The paper's methodological originality is highly limited. The core architecture is not novel but is fundamentally a modification of an existing SOTA diffusion model (Wan2.1) using a standard causal attention mask. The entire framework relies on knowledge distillation, meaning the impressive performance is likely inherited from the powerful teacher model rather than originating from the proposed techniques. This blurs the line of contribution, making it unclear whether this paper presents a superior autoregressive paradigm or merely a successful, but non-novel, distillation recipe for an existing model.
2.Grandiose Claims Built on Critically Insufficient Evidence: The paper's most ambitious and impactful claim—its capability for zero-shot long-video generation—is supported by shockingly thin evidence. The entire proof rests on a single, highly convenient qualitative example (Figure 6, a jellyfish scene) presented as static frames.
a) No Supplementary Material: For a video generation paper, the lack of supplementary video materials is unacceptable. It makes one of its central claims entirely unverifiable by the reviewers, as temporal consistency, motion quality, and potential artifacts cannot be assessed from static images.
b) Cherry-Picking Risk: The chosen example is an ideal case that minimizes the need for long-term coherence. This strongly suggests cherry-picking and fails to demonstrate robustness on more challenging, narrative-driven content.
3. Marginal and Statistically Questionable Performance Gains: The numerical improvements reported are too marginal to substantiate the paper's strong claims of superiority.
a) vs. Base Model: The gain over its own teacher model (Wan2.1) is a mere +0.99 on the VBench Total Score (84.87 vs. 83.88). Without any statistical significance testing, this could very well be within the range of measurement noise.
b) Within Ablations: The ablation studies reveal the low efficacy of the core methodological proposals. The complex Symmetric DMD offers only a +0.31 improvement over a much simpler causal scorer (Table 4). These minimal gains fail to justify the added complexity and the paper's narrative about the transformative impact of its methods.
4. Arbitrary and Under-justified Methodological Choices: The "spatiotemporal cube" paradigm, presented as a principled alternative to frames, rests on an arbitrary and unjustified causal ordering. The implicit raster-scan order (time -> height -> width) is unnatural and is never defended against other possible orderings. Furthermore, ablation results (Table 3) show that performance is highly sensitive to the cube size, suggesting the "cube" is not a fundamental unit but rather a fragile, fine-tuned hyperparameter, which undermines the paper's core philosophical argument.

**Questions:**

1. Your core generator is a modified version of a pre-existing SOTA model (Wan2.1) trained via distillation. Could you clarify what you see as the primary novel contribution of this work? How can you disentangle the performance gains from the powerful inductive biases already present in the teacher model versus the actual effectiveness of your proposed cube-based factorization and Symmetric DMD?
2. The extraordinary claim of generating minute-long videos is supported only by a single static figure (Figure 6) without any supplementary videos. This is insufficient for a proper evaluation of temporal dynamics. Can you justify why supplementary materials were not provided? And can you provide more diverse and challenging examples (e.g., with consistent characters or narratives) to substantiate this claim and address the risk of semantic drift due to the fixed-size KV cache?
3. The VBench score improvement over your teacher model (Wan2.1) is marginal (+0.99), and the gain from your complex Symmetric DMD over a simple causal scorer is even smaller (+0.31). How do you justify the significance of your method when the empirical gains are so minimal and potentially not statistically significant? Does this not suggest that the primary real-world advantage of your model is its speed, rather than a fundamental improvement in generation quality or a new paradigm?
4. The "spatiotemporal cube" approach relies on a specific, yet undefended, raster-scan ordering (time -> height -> width). Could you provide a principled reason for this arbitrary choice over other possible causal orderings? Furthermore, given that performance is highly sensitive to the cube's dimensions (Table 3), how does this reconcile with the claim that cubes are a more "natural" or fundamental prediction unit than frames?

---

### Official Review · Reviewer_gVEE · 2025-11-01

**Soundness:** 3
**Presentation:** 3
**Contribution:** 3
**Rating:** 6
**Confidence:** 4

**Summary:**

The paper proposes VideoAR, a framework that supports various prediction units: full frames, key-detail frames, multiscale refinements, and spatiotemporal cubes. Among these, the authors favor spatiotemporal cubes, which enable autoregressive modeling while eliminating the rigid frame-by-frame generation constraint. Experiments show that cube-based prediction consistently achieves higher generation quality, faster inference, and better temporal coherence, outperforming state-of-the-art methods on VBench and enabling stable generation of minute-long videos.

**Strengths:**

- VideoAR systematically explores four autoregressive units for video generation.
- VideoAR proposes SymmetryDMD to resolve the mismatch between score functions.
- VideoAR achieves state-of-the-art performance and faster inference via causal attention and distillation.

**Weaknesses:**

- **Marginal improvements across chunk-wise units.** The key-detail (frame chunks), scale (scale chunks), and cube (fine-grained frame chunks) configurations all achieve strong VBench performance (≥84.2). For simplicity, I consider CausVid [1] / MAGI-1 [2]’s frame-chunk formulation sufficient for autoregressive generation. The authors should include CausVid’s frame-chunk results as a baseline and provide a detailed analysis of whether the more complex cube-chunk formulation is truly necessary.
- **Comparsion with AsymmetricDMD.** VideoAR introduces SymmetryDMD, a distillation method that bridges bidirectional and causal attention patterns. The authors should compare and analyze this approach against CausVid’s AsymmetricDMD to clearly demonstrate its advantages and improvements. In particular, does performance gain under the frame unit (83.72 vs. 81.46) primarily from this distillation difference?

[1] Tianwei Yin, Qiang Zhang, Richard Zhang, William T. Freeman, Fredo Durand, Eli Shechtman, Xun Huang. From Slow Bidirectional to Fast Causal Video Generators. In CVPR, 2025.

[2] Hansi Teng, Hongyu Jia, Lei Sun, Lingzhi Li, Maolin Li, Mingqiu Tang, Shuai Han, Tianning Zhang, WQ Zhang, Weifeng Luo, et al. Magi-1: Autoregressive video generation at scale. arXiv:2505.13211, 2025.

**Questions:**

- See weaknesses.
- In short, the work introduces the VideoAR framework for autoregressive video generation. The contributions of this work are substantiated by the experimental results. However, to convincingly demonstrate that VideoAR represents a genuine improvement over CausVid and Self-Forcing [3], the design of its unit architecture and DMD module should be compared in greater detail with those methods.

[3] Xun Huang, Zhengqi Li, Guande He, Mingyuan Zhou, Eli Shechtman. Self Forcing: Bridging the Train-Test Gap in Autoregressive Video Diffusion. arXiv:2506.08009, 2025.

---

### Official Review · Reviewer_B4ZT · 2025-11-01

**Soundness:** 3
**Presentation:** 4
**Contribution:** 2
**Rating:** 4
**Confidence:** 5

**Summary:**

This paper proposes VideoAR with two main contributions. First, it ablates different video component for video autoregressive generation, including frame, key-detail frame, cube, and multi-scale. Second, it proposes Symmetric DMD, to distill a full-attention video generation model into a uni-directional generator, which generates each unit in one-step. VideoAR are tested on VBench with superior performance.

**Strengths:**

1. The problem of prediction units is clear and well formulated. The paper challenges the default “frame = token” assumption and formalizes AR video generation over arbitrary prediction units (frames, key–detail groups, multiscale, spatiotemporal cubes).
2. Symmetric DMD is motivated and validated. The proposed Symmetric DMD successfully convert a teacher model to a causal generator, with no performance loss.
3. The final model achieves superior performance on VBench while being significantly faster with the help of KV cache.

**Weaknesses:**

1. My biggest concern is that the model heavily relies on a well-trained teacher model. The performance of the model mainly comes from the teacher model (84.87 v.s. 84.34). The Symmetric DMD cannot be applied directly on raw data. The framework is more of a one-step unidirectional distillation instead of a general AR framework. Distilling the teacher model to a few-step diffusion with DMD will be much faster.
2. The study on different prediction units is superficial. Ablation in Table 2 does not really show much details about the different units. For example, are they of the same length? Are the hyperparameters in other settings ablated? Without this, it is hard to decide that cube is the best option.
3. Long video are mostly qualitative. Methods like FIFO could easily extends to longer context. Given wan is the teacher model, I highly suspect AR will show any ability beyond it.
4. The gain is not clear. What kind of internal dataset is used? What prompts are used to sample from Wan? Are they curated based on VBench? Without those information, the performance gain via distillation is questionable.

**Questions:**

Please refer to weaknesses.

---

### Official Review · Reviewer_7sNJ · 2025-11-01

**Soundness:** 2
**Presentation:** 2
**Contribution:** 2
**Rating:** 4
**Confidence:** 4

**Summary:**

This paper introduces VideoAR, an autoregressive framework for video generation that challenges the conventional frame-by-frame prediction paradigm. The authors propose to decompose video into more flexible "prediction units", such as full frames, key-detail frames, multi-scale pyramids, and spatiotemporal cubes. The paper finds that using spatiotemporal cubes, which allows the model to predict across both spatial and temporal dimensions simultaneously, consistently yields the best performance. Furthermore, it maintains high inference speed and demonstrates a remarkable capability for zero-shot long video generation.

**Strengths:**

1. The paper is well-written. The proposed method is well-illustrated and easy to follow.

2. **Thorough ablation studies:** The authors have conducted a comprehensive set of ablation studies that validate their design choices. The comparison of different prediction units (Table 2) clearly demonstrates the superiority of the cube-based approach. Furthermore, the ablations on cube size (Table 3) and the components of the Symmetric DMD loss (Table 4) add significant weight to the paper's conclusions.

**Weaknesses:**

1. **Reduced novelty due to prior work:** The claim of generalizing the prediction unit beyond the frame-level is not entirely novel. Prior work, specifically "Next Block Prediction: Video Generation via Semi-Autoregressive Modeling", has already explored a similar concept of semi-autoregressive modeling using block-wise predictions. This existing research lessens the originality of the core contribution. The authors should explicitly discuss and differentiate their approach from NBP to better situate their work within the existing literature.

2. **Lack of qualitative comparison for prediction units:** While the ablation study in Table 2 quantitatively demonstrates that the `2x2x2` spatiotemporal cube is the most effective prediction unit, the paper lacks qualitative comparisons to support this finding. There are no visual examples (e.g., side-by-side video clips) that illustrate the differences in generation quality, motion coherence, or error accumulation between videos generated with frames, key-detail frames, and cubes. Such a comparison would provide valuable intuition and a more concrete understanding of *why* the cube-based approach is superior.

3. **Details on long video generation:** The paper mentions "gradually drops earlier KV cache to keep the overall sequence length short," but lacks specifics. Details on the cache management policy (e.g., is it a fixed-size sliding window?), the context length used in experiments, and an analysis of the trade-off between context length, temporal coherence, and computational cost would be valuable additions.

**Questions:**

1. **On symmetric DMD:** Could you provide more intuition on why the combination of a forward and a backward causal scorer is a suitable proxy for a single bidirectional scorer in terms of modeling capacity? Does this architectural choice introduce any specific inductive biases during training compared to using a single, non-causal scorer for s_gen as in the original DMD?

---

### Note · Authors · 2025-11-13

I have read and agree with the venue's withdrawal policy on behalf of myself and my co-authors.